# Morpho-Molecular Characterization of Microfungi Associated with *Phyllostachys* (Poaceae) in Sichuan, China

**DOI:** 10.3390/jof8070702

**Published:** 2022-07-01

**Authors:** Qian Zeng, Yi-Cong Lv, Xiu-Lan Xu, Yu Deng, Fei-Hu Wang, Si-Yi Liu, Li-Juan Liu, Chun-Lin Yang, Ying-Gao Liu

**Affiliations:** 1College of Forestry, Sichuan Agricultural University, Chengdu 611130, China; zq1573037145@163.com (Q.Z.); yiconglv0616@163.com (Y.-C.L.); xuxiulanxxl@126.com (X.-L.X.); dy17711433642@126.com (Y.D.); scbzwfh@163.com (F.-H.W.); sharonwinchester@126.com (S.-Y.L.); liulijuan429@163.com (L.-J.L.); 2Forestry Research Institute, Chengdu Academy of Agricultural and Forestry Sciences, Chengdu 611130, China

**Keywords:** bambusicolous fungi, molecular phylogeny, one new genus, systematics, two new species

## Abstract

In the present study, we surveyed the ascomycetes from bamboo of *Phyllostachys* across Sichuan Province, China. A biphasic approach based on morphological characteristics and multigene phylogeny confirmed seven species, including one new genus, two new species, and five new host record species. A novel genus *Paralloneottiosporina* is introduced to accommodate *Pa. sichuanensis* that was collected from leaves of *Phyllostachys violascens*. Moreover, the newly introduced species *Bifusisporella sichuanensis* was isolated from leaves of *P. edulis*, and five species were newly recorded on bamboos, four species belonging to *Apiospora*, viz. *Ap. yunnana*, *Ap. neosubglobosa*, *Ap. jiangxiensis*, and *Ap. hydei*, and the last species, *Seriascoma yunnanense,* isolated from dead culms of *P. heterocycla*. Morphologically similar and phylogenetically related taxa were compared. Comprehensive descriptions, color photo plates of micromorphology are provided.

## 1. Introduction

Bamboo is currently classified in the subfamily Bambusoideae of the extensive grass family Poaceae, and distributed worldwide. It comprises circa 1000 to 1500 species in up to 90 genera [1] and more than 70 species in *Phyllostachys* (Bambusoideae, Poaceae) [2,3]. Most bamboos are distributed in Southeast Asia, with China as the distribution center [4]. There are about 21 species of *Phyllostachys* in Sichuan, including *Phyllostachys edulis* (Carriere) J. Houzea, *P. heteroclada* Oliver, and *P. violascens* ‘Prevernalis’ S.Y. Chen et C.Y. Yao. Bamboos of *Phyllostachys* play an important role in native economy and ecology. They are used in furniture, and construction (e.g., fishing rods, flutes, flooring materials, chairs.) [5,6]. Bamboo shoots are used as food for humans and animals such as pandas [7,8]. In addition, it is an important ornamental plant for the landscape in China because of its evergreen and graceful appearance [9].

A review of the literature on bamboo-associated fungi reveals that nearly 1500 species have been described or recorded worldwide [10], including economically important pathogenic fungi, and a large number of saprobic and endophytic fungi [1,11,12,13]. Most bambusicolous fungi have been reported from Asia, especially Japan and Thailand, a few known from India and South America [1,12,14,15,16,17,18]. However, few studies have investigated the diversity and phylogeny on bamboo in China. The taxonomic studies on bambusicolous fungi are of great significance [19,20,21]. According to the literature review, about 85 species associated with *Phyllostachys* have been recorded. Teng [22] first reported the fungus *Oedocephalum glomerulosum* (Bull.) Sacc. on *Phyllostachys* in 1932. Tai listed 36 species of *Phyllostachys* from bamboo based on the reports on Chinese fungal resource until 1973 [23]. Chen investigated the phytogeography of forest fungi in China, North America, and Siberia, from which 33 species were found associated with *Phyllostachys* [24]. However, most of those identifications were conducted lacking molecular data and detailed micromorphology, and as most bamboos are unidentified, the relationship of bambusicolous fungi with bamboo species is not clear.

Due to the high fungal diversity on *Phyllostachys*, an ongoing investigation was conducted in several main producing or planting areas of bamboo *Phyllostachys* in Sichuan Province, China, including Ya’an City, Qionglai City, Chengdu City, and Yibin City. In this study, we provide detailed taxonomic features combining morphology and phylogeny on the fungi associated with *Phyllostachys* from Sichuan Province, China, which is a fundamental task for the bioresource collection on bambusicolous fungi.

## 2. Materials and Methods

### 2.1. Specimen Collection and Morphological Study

From 2020 to 2021, the specimens were collected from leaves, branches, and culms. The samples were kept in plastic bags and taken back to the laboratory after being photographed with a Sony DSC-HX3 digital camera. The fungi were isolated into pure culture based on single spore isolation [25]. Glass slide specimens were prepared by free-hand slicing with double-sided blades for morphologic observation. Morphological characteristics of ascomata and sporodochia were observed using a dissecting microscope, the NVT-GG (Shanghai Advanced Photoelectric Technology Co. Ltd., Shanghai, China), and photographed with a VS-800C micro-digital camera (Shenzhen Weishen Times Technology Co. Ltd., Shenzhen, China). An Olympus BX43 compound microscope with an Olympus DP22 digital camera was used to observe and photograph the microstructure of asci, ascospores, conidiophores, and conidia. Measurements were performed using Tarosoft^®^ Image Frame Work v.0.9.7 (Tarosoft (R), Nontha Buri, Thailand). Specimens were deposited at the Herbarium of Sichuan Agricultural University, Chengdu, China (SICAU), and pure cultures were deposited at the Culture Collection in Sichuan Agricultural University (SICAUCC).

### 2.2. DNA Extraction, PCR Amplification, and Nucleotide Sequencing

Genomic DNA was extracted from fresh mycelia which was cultured on PDA at 25 °C for 15–30 days, using a Trelief^TM^ Plant Genomic DNA Kit. Primers ITS5/ITS4 [26], NS1/NS4 [26], LR0R/LR5 [27], T1/Bt2b [28,29], RPB1-Ac/RPB1-Cr [30,31], and fRPB2-5F/fRPB2-7cR [32] were used for the amplification of internal transcribed spacers (ITS), the partial small subunit nuclear rDNA (SSU), the partial large subunit nuclear rDNA (LSU), the β-tubulin gene (*tub2*), the large subunit of RNA polymerase I (*rpb1*), and RNA polymerase II second largest subunit (*rpb2*) genes, respectively. Primers EF1-983F/EF1-2218R [33] and EF1-728F/EF2 [34,35] were employed for translation elongation factor 1-alpha (*tef1-α*) genes.

Amplification reactions were performed in 25 µL of total reaction that contained 22 µL Master Mix (Beijing TsingKe Biotech Co., Ltd., Beijing, China), 1 µL each of forward and reverse (10 µM) primers and 1 µL of DNA template. The amplification reactions were performed as described by Dai et al. [16] and Wang et al. [36]. PCR products were purified and sequenced at TsingKe Biological Technology Co., Ltd. (Chengdu, China). The resulting sequences were submitted to GenBank.

### 2.3. Sequence Alignment and Phylogenetic Analyses

Based on blast searches in GenBank, using ITS, LSU, SSU, *tef1-α*, *tub2*, *rpb1,* or *rpb2* sequence data, separate phylogenetic analyses were carried out to determine the placements of each fungal group (Table 1). Sequences for phylogenetic analyses were selected mainly from recently published literature and phylogenetic related sequences based on BLAST searches in GenBank (Table A1). Datasets were aligned using MAFFT v.7.407 [37], and ambiguous regions were excluded with BioEdit version 7.0.5.3 [38]. Maximum likelihood (ML) and Bayesian inference (BI) were constructed as described in Xu et al. [39]. The phylogram was visualized with FigureTree v. 1.4.3 and edited using Adobe Illustrator CS6 (Adobe Systems Inc., San Jose, CA, USA).

## 3. Results

### 3.1. Phylogenetic Analyses

A combined dataset (ITS, LSU, *tef1-α*, *tub2*) comprising 138 taxa within Apiosporaceae, which is rooted with *Pestalotiopsis chamaeropis* (CBS 237.38) and *Pe. colombiensis* (CBS 118553) (Pestalotiopsidaceae, Amphisphaeriales), was used for the phylogenetic analyses. The alignment contained 5875 characters (ITS = 999, LSU = 1382, *tef1-α* = 1651, *tub2* = 1844), including gaps. The best scoring RAxML tree with a final likelihood value of −36198.939448 is presented. The matrix had 2337 distinct alignment patterns, with 64.85% of undetermined characters or gaps. Estimated base frequencies were as follows: A = 0.237208, C = 0.257370, G = 0.253511, T = 0.251911, with substitution rates AC = 1.104968, AG = 2.746651, AT = 1.143208, CG = 0.910079, CT = 4.335389, GT = 1.000000. The gamma distribution shape parameter α = 0.269105, and the tree length = 3.509694. In the phylogenetic trees generated from ML and BI analyses, the strain SICAUCC 22-0032 clustered with the known species *Apiospora hydei* (KUMCC 16-0204, CBS 114990) in a clade with 97% ML and 0.99 BYPP support value, strain SICAUCC 22-0070 clustered with *Ap. jiangxiensis* (CGMCC 3.18381, LC4578) with high support values (100% ML and 1.00 BYPP), strain SICAUCC 22-0071 clustered with *Ap. neosubglobosa* (JHB006, JHB007) in a clade with 100% ML and 1.00 BYPP support value, and strain SICAUCC 22-0072 clustered with the *Ap. yunnana* (MFLUCC 15-0002) in a clade with 100% ML and 1.00 BYPP support values (Figure 1).

Phylogenetic analyses of a concatenated aligned dataset (ITS, LSU, *rpb1*, *tef1-α*), including 70 taxa within Magnaporthaceae and Pyriculariaceae, were conducted and rooted with *Ophioceras dolichostomum* (CBS 114926) and *O. leptosporum* (CBS 894.70) (Ophioceraceae, Magnaporthales). The alignment contained 4094 characters (ITS = 899, LSU = 1105, *rpb1* = 1047, *tef1-α* = 1043), including gaps. The best scoring RAxML tree with a final likelihood value of −31022.648763 is presented. The matrix had 1923 distinct alignment patterns, with 36.77% of undetermined characters or gaps. Estimated base frequencies were as follows: A = 0.243596, C = 0.275654, G = 0.281915, T = 0.198836, with substitution rates AC = 1.103727, AG = 2.292134, AT = 1.431191, CG = 0.918700, CT = 5.773674, GT = 1.000000. The gamma distribution shape parameter α = 0.319184, and the tree length = 3.313974. In the phylogenetic tree (Figure 2), the novel species *Bifusisporella sichuanensis* constitutes a highly supported independent lineage (ML = 100%, BYPP = 1.00) with *B. sorghi* (URM 7864, URM 7442).

The concatenated aligned dataset of ITS, LSU, SSU, *tef1-α* sequences, including 124 ingroup taxa within Phaeosphaeriaceae and two outgroup taxa in Leptosphaeriaceae, were used for the phylogenetic analyses of *Paralloneottiosporina*. The alignment contained 5851 characters (ITS = 1469, LSU = 1433, SSU = 1548, *tef1-α* = 1401), including gaps. The best scoring RAxML tree with a final likelihood value of −46908.078740 is presented. The matrix had 2382 distinct alignment patterns, with 55.68% of undetermined characters or gaps. Estimated base frequencies were as follows: A = 0.246158, C = 0.236637, G = 0.264322, T = 0.252883, with substitution rates AC = 1.087661, AG = 2.657942, AT = 2.045792, CG = 0.863381, CT = 6.106747, GT = 1.000000. The gamma distribution shape parameter α = 0.263651, and the tree length = 7.503091. In the phylogenetic tree generated from ML and BI analyses, the novel species *Paralloneottiosporina sichuanensis* (SICAUCC 22-0074, SICAUCC 22-0075) constitutes a moderately supported independent lineage (63% ML/0.99 BYPP statistical support) with the species *Alloneottiosporina thailandica* (MFLUCC 15-0576) (Figure 3).

A combined dataset (ITS, LSU, SSU, *tef1-α*, *rpb2*) comprising 25 taxa within Bambusicolaceae, Biatriosporaceae, Roussoellaceae, Torulaceae, and Paradictyoarthriniaceae was used for phylogenetic analyses of *Seriascoma*, and the *Westerdykella ornata* (CBS 379.55) (Sporormiaceae) was used as outgroup taxon. The alignment contained 6569 characters (LSU = 1383, SSU = 1741, *tef1-α* = 1346, *rpb2* = 2099), including gaps. The best scoring RAxML tree with a final likelihood value of −22606.776997 is presented. The matrix had 1406 distinct alignment patterns, with 48.40% of undetermined characters or gaps. Estimated base frequencies were as follows: A = 0.250203, C = 0.247742, G = 0.269455, T = 0.232600, with substitution rates AC = 1.348170, AG = 4.119625, AT = 1.278817, CG = 1.296090, CT = 9.080955, GT = 1.000000. The gamma distribution shape parameter α = 0.146142, and the tree length = 1.192279. According to the phylogenetic tree (Figure 4), the strain (SICAUCC 22-0059) clustered with *Seriascoma yunnanense* (MFLU 19-0690) in a clade with 100% ML and 1.00 BYPP statistical support.

### 3.2. Taxonomy

Apiosporaceae K.D. Hyde, J. Fröhl., Joanne E. Taylor & M.E. Barr, Sydowia. 50 (1): 23 (1998).

*Apiospora hydei* (Crous) Pintos & P. Alvarado, Fungal Systematics and Evolution. 7: 206 (2021) (Figure 5).

≡ *Arthrinium hydei* Crous, IMA Fungus 4(1): 142 (2013).

Saprobic on dead culms of *Phyllostachys nigra* (Lodd. ex Lindl.) Munro. *Sexual morph*: *Ascostromata* 421–1343 × 174–387 × 176–245 μm ($\bar{x}$ = 705 × 267 × 198 μm, *n* = 30), solitary to gregarious, immersed, fusiform to ellipsoid, dark brown to black, multi-loculate, with long axis. *Peridium* 17–46 μm wide, composed of 8–15 layers of brown to hyaline cells of *textura angularis* to *prismatica*. *Hamathecium* 2–6.5 μm wide, composed of dense, long, septate, and unbranched paraphyses. *Asci* 81–123 × 16–23 μm, ($\bar{x}$ = 116 × 180 μm, *n* = 50), 8–spored, unitunicate, broadly cylindrical, slightly curved, with a short pedicel, apically rounded. *Ascospores* 24–30 × 7–11 μm, ($\bar{x}$ = 26 × 10 μm, *n* = 50), 2-seriate, elliptical, 1–septate, with a large, curved upper cell and small lower cell, with narrowly rounded ends, hyaline, guttules, smooth-walled, surrounded by gelatinous sheath. *Asexual morph*: see Crous et al. [40].

Material examined: China, Sichuan Province, Chengdu City, Wenjiang District (19°30′42.22″ N, 103°51′19″ E, Alt. 528 m), on dead culms of *Phyllostachys nigra*, 14 March 2021, Yi-cong Lv, LYC202103003 (SICAU 22-0032), living culture SICAUCC 22-0032.

Culture characters: Ascospores germinate within 24 h. Colonies grow fast on PDA, reaching 6 cm after one week at 25 °C, under 12 h light/12 h dark, and are cottony, circular, and white from above and light yellow below, with irregular edge.

Notes: *Apiospora hydei* was introduced based on the asexual morph characters and phylogeny analyses by Crous et al. [40]. Morphological comparisons were impossible due to the lack of sexual morph between our isolates and the ex-type strain (CBS 114990), but it is similar to *A. hydei* in sexual descriptions provided by Dai et al. [41]. Nucleotide comparisons of ITS, LSU, *tef1-α* and *tub2* (SICAUCC 22-0033) showed high homology with the sequences of *A. hydei* (CBS 114990), similarities are 100% (528/528, 0 gaps), 99.77% (896/898, 0 gaps), 99.71% (355/356, 0 gaps), and 98.82% (754/763, 0 gaps), respectively.

*Apiospora jiangxiensis* (M. Wang & L. Cai) Pintos & P. Alvarado, Fungal Systematics and Evolution 7: 206 (2021) (Figure 6).

≡ *Arthrinium jiangxiense* M. Wang & L. Cai, in Wang, Tan, Liu & Cai, MycoKeys 34(1): 14 (2018).

Saprobic on dead culms of *Phyllostachys heteroclada* Oliver. *Sexual morph*: *Ascostromata* 575–1334 × 274–444 × 134–157 μm ($\bar{x}$ = 876 × 355 × 143 μm, *n* = 30), solitary to gregarious, multi-loculate, immersed, fusiform to ellipsoid, black, with long axis broken at the top. *Peridium* 9.0–44 μm wide ($\bar{x}$ = 21 μm, *n* = 25), composed of several layers of brown to hyaline cells of *textura angularis* to *prismatica*. *Hamathecium* 4.0–11 μm wide, composed of dense, long, septate, unbranched, paraphyses. *Asci* 83–114 × 18–28 μm ($\bar{x}$ = 104 × 23 μm, *n* = 50), 8–spored, unitunicate, broadly cylindrical to long clavate, with a short pedicel, slightly curved, apically rounded. *Ascospores* 32–37 × 9.6–11 μm ($\bar{x}$ = 34 × 10 μm, *n* = 50), 2–seriate, 1–septate, elliptical, with a large, curved, upper cell and small lower cell, with narrowly rounded ends, hyaline, smooth-walled, with many guttules, surrounded by gelatinous sheath attached. *Asexual morph*: see Wang et al. [36].

Material examined: China, Sichuan Province, Luzhou City, Xuyong District (27°53′28″ N, 105°16′36″ E, Alt. 1350 m), on dead culm of *Phyllostachys heteroclada*, 26 July 2021, Qian Zeng, ZQ202107133 (SICAU 22-0070), living culture SICAUCC 22-0070.

Culture characters: Ascospores germinate on PDA within 24 h. Colonies grow fast on PDA, reaching 6 cm after 1 week at 25 °C, under 12 h light/12 h dark, and are cottony, white, circular, with irregular edge.

Notes: Specimen in our study shared similar morphology with the original description of *Apiospora jiangxiensis* by Wang et al. [36]. Nucleotide comparisons of ITS, LSU, and *tub2* (SICAUCC 22-0070) showed high homology with the sequences of *Ap. jiangxiensis* (CGMCC 3.18381), similarities are 100% (541/541, 0 gaps), 99.09% (436/440, 0 gaps), and 98.22% (717/730, 0 gaps), respectively. However, the latter lack *tef1-α* sequences for further comparisons.

*Apiospora neosubglobosa* (D.Q. Dai & H.B. Jiang) Pintos & P. Alvarado, Fungal Systematics and Evolution 7: 206 (2021) (Figure 7).

≡ *Arthrinium neosubglobosum* D.Q. Dai & H.B. Jiang, Mycosphere 7(9): 1337 (2017).

Saprobic on dead culms of *Phyllostachys bissetii* McClure. *Sexual morph*: *Ascostromata* 330–1092 × 198–354 × 134–224 μm ($\bar{x}$ = 632 × 250 × 174 μm, *n* = 30), gregarious, immersed, multi-loculate, fusiform to ellipsoid, dark brown to black, with long axis broken at the top. *Peridium* 17.0–46 μm wide ($\bar{x}$ = 19 μm, *n* = 25), composed of several layers of brown to hyaline, cells of *textura angularis* to *prismatica**. Hamathecium* 3.5–6.0 μm wide, composed of dense, long, septate, unbranched, paraphyses. *Asci* 94–137 × 23–40 μm ($\bar{x}$ = 125 × 31 μm, *n* = 50), 8-spored, unitunicate, broadly cylindrical to long clavate, with a short pedicel, slightly curved, apically rounded. *Ascospores* 28–36 × 13–15 μm ($\bar{x}$ = 32 × 14 μm, *n* = 50), 2–seriate, 1–septate, elliptical, with a large, curved, upper cell and small lower cell, with narrowly rounded ends, hyaline, smooth-walled, with many guttules, surrounded by gelatinous sheath attached. *Asexual morph*: see Dai et al. [16].

Material examined: CHINA, Sichuan Province, Luzhou City, Xuyong District (27°52′5″ N, 105°16′23″ E, Alt. 1470 m), on dead culm of *Phyllostachys bissetii*, 26 July 2021, Qian Zeng, ZQ202107128 (SICAU 22-0071), living culture SICAUCC 22-0071.

Cultural characters: Ascospores germinate on PDA within 24 h. Colonies grow fast on PDA, reaching 4 cm after 1 week at 25 °C, under 12 h light/12 h dark, and are cottony, circular, initially white, then brown, with regular edge.

Notes: *Apiospora neosubglobosa* was described by Dai et al. based on the morphological characteristics and molecular phylogeny [16]. Strain SICAUCC 22-0071 clustered with ex–type strain (JHB007) with high bootstrap support (100% ML and 1.00 BYPP). Nucleotide comparisons of ITS and LSU (SICAUCC 22-0071) showed high homology with the sequences of *A**p**. neosubglobosa* (JHB007), similarities are 99.84% (649/650, 0 gaps), 100% (1173/1173, 0 gaps), respectively.

*Apiospora yunnana* (D.Q. Dai & K.D. Hyde) Pintos & P. Alvarado, Fungal Systematics and Evolution 7: 207 (2021) (Figure 8).

≡ *Arthrinium yunnanum* D.Q. Dai & K.D. Hyde, Fungal Diversity 82: 69 (2016).

Saprobic on culms of *Phyllostachys aurea* Carr. ex A. et C. Riv. *Sexual morph*: *Ascostromata* 624–1307 × 253–510 × 165–211 μm ($\bar{x}$ = 892 × 359 × 188 μm, *n* = 30), gregarious, multi-loculate, immersed, fusiform to ellipsoid, black, with long axis broken at the top. *Peridium* 8.5–43 μm wide ($\bar{x}$ = 17 μm, *n* = 25), composed of several layers of brown to hyaline cells of *textura angularis* to *prismatica*. *Hamathecium* 3.5–8.0 μm wide, composed of dense, long, septate, unbranched paraphyses. *Asci* 89–144 × 18–40 μm ($\bar{x}$ = 120 × 32 μm, *n* = 50), 8–spored, unitunicate, broadly cylindrical to long clavate, no pedicel, slightly curved, apically rounded. *Ascospores* 30–42 × 10–13 μm ($\bar{x}$ = 36 × 12 μm, *n* = 50), 2–seriate, 1–septate, elliptical, with a large, curved, upper cell and a small lower cell, with narrowly rounded ends, hyaline, smooth-walled, with many guttules, surrounded by gelatinous sheath attached. *Asexual morph*: see Dai et al. [16].

Material examined: China, Sichuan Province, Yibin City, Changning District (28°28′8″ N, 105°0′16″ E, Alt. 890 m), on dead culm of *Phyllostachys aurea*, 23 July 2021, Qian Zeng, ZQ202107027 (SICAU 22-0072), living culture, SICAUCC 22-0072.

Culture characters: Ascospores germinate on PDA within 24 h and germ tubes produced from sides. Colonies grow fast on PDA, reaching 6 cm after 1 week at 25 °C, under 12 h light/12 h dark, and are cottony, circular, and white with irregular edge.

Notes: The sexual and asexual morph of *Apiospora yunnana* was reported by Dai et al. [16]. Morphologically, our observations were identical to the sexual descriptions provided by Daiet et al. [16]. Nucleotide comparisons of ITS and LSU (SICAUCC 22-0072) showed high homology with the sequences of *Ap. yunnana* (MFLUCC 15-0002), similarities are 99.85% (667/668, 0 gaps), 100% (847/847, 0 gaps), respectively. However, the latter lack *tef1-α* and *tub2* sequences for further comparisons.

Magnaporthales Thongkantha, Vijaykrishna & K.D. Hyde. Fungal Diversity. 34: 157–173 (2009).

Magnaporthaceae P.F. Cannon, Systema Ascomycetum 13: 26 (1994).

*Bifusisporella* R.M.F. Silva, R.J.V. Oliveira, J.D.P. Bezerra, J.L. Bezerra, C.M. Souza-Motta & G.A. Silva, Mycological Progress 18(6): 852 (2019).

Type species: *Bifusisporella sorghi* R.M.F. Silva, R.J.V. Oliveira, J.D.P. Bezerra, J.L. Bezerra, C.M. Souza-Motta & G.A. Silva.

Description: Endophytic and parasitic fungi on Poaceae. *Sexual morph: Ascomata* separate or gregarious, subglobose, black, coriaceous, semi-immersed, unilocular or multilocular. *Peridium* with hyaline to brown cells of *textura angularis*. *Hamathecium* hyaline, with distinct septa, wider at the base, tapering towards the apex. *Asci* 8–spored, cylindrical, with a J-, apical ring, developing from the base and periphery of the ascomata, with a short pedicel. *Ascospores* biseriate, hyaline, fusiform, with distinct septa, with narrowly rounded ends, without appendages. *Asexual morph**:* Found in *Bifusisporella sorghi* cultures by Silva et al. [42].

Notes: *Bifusisporella* was introduced as a new genus to accommodate *B. sorghi* based on morphology and phylogeny. At present, *Bifusisporella* comprises only the ex-type species *B. sorghi*, and no records on its sexual morph. The new species *B. sichuanensis* is well-supported within *Bifusisporella,* which suggests that there is a need to amend the morphological circumscriptions of the genus.

*Bifusisporella sichuanensis* Q. Zeng, Y.C. Lv & C.L. Yang, sp. nov. (Figure 9).

Index Fungorum: IF559625

Etymology: Refers to the region from where the fungus was collected.

Holotype: SICAU 22-0073

Parasitic on living leaves of *Phyllostachys*
*edulis* (Carriere) J. Houzeau. *Sexual morph: Ascostromata* 536–1672 × 332–849 × 125–245 μm ($\bar{x}$ = 1103 × 591 × 193 μm, *n* = 30), separate or gregarious, subglobose, black, coriaceous, semi-immersed, unilocular or multilocular, glabrous. *Peridium* 14–34 μm wide ($\bar{x}$ = 20 μm, *n* = 30), composed of 3–9 layers, with hyaline to brown cells of *textura angularis*. *Hamathecium*, hyaline, cellular, with distinct septa. *Asci* 79–126 × 9.5–13 μm ($\bar{x}$ = 99 × 11 μm, *n* = 30), 8–spored, bitunicate, cylindrical, with an apical chamber and a short pedicel. *Ascospores* 22–35 × 5.0–6.5 μm ($\bar{x}$ = 29 × 5.5 μm, *n* = 50), overlapping, biseriate, hyaline, fusiform, 3–septate, rarely constricted at septate, with narrowly rounded ends, smooth-walled, guttules, without gelatinous sheath. *Asexual morph*: Undetermined.

Material examined: China, Sichuan Province, Yibin City, Xingwen District (28°15′22″ N, 105°6′29″ E, Alt. 850 m), on living to nearly dead leaves of *Phyllostachys edulis*, 25 July 2021, Qian Zeng, ZQ202107111 (SICAU 22-0073 holotype), ex-type living culture, SICAUCC 22-0073.

Culture characters: Ascospores germinate in sterilized water within 12 h at 25 °C. Colonies grow slow on PDA, reaching approximately 2 cm in 30 days at 25 °C, under 12 h light/12 h dark, and are irregular, black, frilly with white margin, and black on the back of colonies.

Notes: *Bifusisporella sichuanensis* is phylogenetically close (100% ML and 1.00 BYPP) to *B. sorghi* (URM 7442) introduced by Silva et al. [42], which is described with asexual morph. However, striking base-pair differences are noted, viz. 11.43% (55/481, 0 gaps), 3.36% (27/803, 0 gaps), 5.11% (24/469, 0 gaps), 9.04% (64/708, 0 gaps) in the ITS, LSU, *tef1-α* and *rpb1*, respectively. Hence, our collection is proposed as a new species.

Pleosporales Luttr. ex M.E. Barr, Prodromus to class Loculoascomycetes: 67 (1987).

Phaeosphaeriaceae M.E. Barr, Mycologia 71: 948 (1979).

*Paralloneottiosporina* Q. Zeng, Y.C. Lv & C.L. Yang, gen. nov.

Index Fungorum: IF559626.

Type species: *Paralloneottiosporina sichuanensis* Q. Zeng, Y.C. Lv & C.L. Yang.

Etymology: Name reflects the morphological similarity to the genus *Alloneottiosporina*.

Parasitic on living to nearly dead leaves of *Phyllostachys violascens* ‘Prevernalis’ S.Y. Chen et C.Y. Yao. *Sexual morph*: *Ascomata* visible as raised to superficial on host, gregarious, globose to subglobose or dome shape, dark brown to black, unilocular, glabrous. *Ostiole* single, circular, centrally located. *Peridium* multi-layered, brown to dark brown cells of *textura angularis*. *Hamathecium* hyaline, numerous, septate, often constricted at septa. *Asci* 8-spored, bitunicate, rounded at apex, cylindrical, curved, with a short pedicel. *Ascospores* hyaline, fusiform, 1–2 septate, constricted at the septum, guttules, smooth-walled, with narrowly rounded ends. *Asexual morph*: *Conidiomata* brown to dark brown, globose to long ellipsoid, coriaceous, semi-immersed, unilocular, gregarious, glabrous. *Conidiomatal wall* comprising multi-layered, dark brown to black cells of *textura angularis*. *Conidia* ellipsoid to ovoid, 1–septate, slightly constricted at the septum, smooth-walled, hyaline, with a rounded apex and a truncated base, guttules.

Notes: *Paralloneottiosporina* resembles *Alloneottiosporina* in asexual status having semi-immersed, unilocular, gregarious, glabrous conidiomata, but *Paralloneottiosporina* differs in absent of microconidia, conidia without mucoid appendages, bigger conidia, fewer layers of conidiomatal wall. The macroconidia of *Alloneottiosporina* species are usually accompanied with mucoid appendages at both ends, and microconidia are produced near the ostiolar channel. Moreover, colonies are whitish to bright orange-pink on PDA in *Paralloneottiosporina*, but olivaceous-black in *Alloneottiosporina* [43]. Based on morphological characteristics and molecular phylogeny, the new genus is introduced in Phaeosphaeriaceae.

*Paralloneottiosporina sichuanensis* Q. Zeng, Y.C. Lv & C.L. Yang, sp. nov. (Figure 10 and Figure 11).

Index Fungorum: IF559627.

Etymology: In reference to Sichuan Province where the specimens were collected.

Holotype: SICAU 22-0074.

Associated with leaf blight on living to nearly dead leaves of *Phyllostachys violascens* (Poaceae). *Sexual morph*: *Ascomata* 106–343 × 39–196 × 55–112 μm ($\bar{x}$ = 168 × 111 × 89 μm, *n* = 30), separate, gregarious to confluent, globose to subglobose, dark brown to black, superficial, unilocular, glabrous. *Ostiole* single, circular, centrally located. *Peridium* 17–38 μm wide ($\bar{x}$ = 29 μm, *n* = 30), composed of 7–12 layers, with brown cells of *textura angularis*. *Hamathecium* hyaline, dense, cellular, with distinct septa. *Asci* 49–97 × 8.5–19 μm ($\bar{x}$ = 71 × 13 μm, *n* = 30), 8-spored, bitunicate, cylindrical, curved, with a short pedicel. *Ascospores* 15–21 × 5.0–7.5 μm ($\bar{x}$ = 18 × 6.0 μm, *n* = 50), overlapping biseriate, straight, hyaline, fusiform, 1–2 septate, constricted at the septum, smooth-walled, with narrowly rounded ends. *Asexual morph: Conidiomata* 90–191 × 61–132 × 81–123 μm ($\bar{x}$ = 132 × 102 × 105 μm, *n* = 30), globose to long ellipsoid, coriaceous, semi-immersed, black, unilocular, gregarious, glabrous. *Conidiomatal wall* 7.5–21 μm wide ($\bar{x}$ = 13 μm), comprising 3–6 layers, brown cells of *textura angularis*. *Conidiophores* reduced to conidiogenous cells. *Conidiogenous cell* 3.0–6.5 × 2.5–5.0 μm ($\bar{x}$ = 5.0 × 3.5 μm, *n* = 20), hyaline, ampulliform to subcylindrical, smooth. *Conidia* 11–20 × 4.0–6.5 μm ($\bar{x}$ = 17 ×5.0 μm, *n* = 50), ellipsoid to ovoid, 1–septate, slightly constricted at the septum, smooth-walled, hyaline, with a rounded apex and a truncated base.

Material examined: China, Sichuan Province, Ya’an City, Yucheng District (29°56′49.54″ N, 102°56′46.03″ E, Alt. 807 m), on living to nearly dead leaves of *Phyllostachys violascens*, 13 May 2020, Qian Zeng, ZQ202005002 (SICAU 22-0074, holotype), ex-type living culture, SICAUCC 22-0074; CHINA, Sichuan Province, Qionglai City, Linjiang Town (30°19′4.42″ N, 103°17′23.06″ E, Alt. 518 m), on living leaves of *Ph. violascens*, 8 November 2020, Qian Zeng, ZQ202011012 (SICAU 22-0075, paratype), living culture, SICAUCC 22-0075.

Culture characteristics: Ascospores germinate in sterilized water within 24 h at 25 °C. Colonies grow slow on PDA, reaching approximately 2.5 cm in 30 days at 25 °C, circular, white aerial mycelium, whitish to bright orange-pink on the surface, and brown on the back.

Pleosporales Luttr. ex M.E. Barr, Prodromus to class Loculoascomycetes: 67 (1987).

Bambusicolaceae D.Q. Dai & K.D. Hyde, Fungal Diversity. 63 (1): 49 (2013).

*Seriascoma yunnanense* Rathnayaka & K.D. Hyde, Asian Journal of Mycology 2(1): 250 (2019) (Figure 12).

Saprobic on dead culm of *Phyllostachys edulis* (Carriere) J. Houzeau. *Sexual morph*: *Ascostromata* 110–200 × 120–150 × 120–140 μm ($\bar{x}$ = 160 × 140 × 130 μm, *n* = 20), solitary to gregarious, immersed, globose to subglobose, coriaceous, dark brown to black. *Peridium* 12–26 μm wide ($\bar{x}$ = 4.0 μm, *n* = 20), composed of 4–9 layers of brown to hyaline cells of *textura angularis*. *Hamathecium* 1.5–2.0 μm wide, composed of dense, branched, long, septate. *Asci* 52–80 × 12–16 μm, ($\bar{x}$ = 60 × 14 μm, *n* = 50), 8-spored, bitunicate, broadly cylindrical, with a short pedicel, straight or slightly curved, with an apical chamber. *Ascospores* 20–30 × 6.0–7.5μm ($\bar{x}$ = 23 × 7.0 μm, *n* = 50), 2–seriate, 1–septate, slightly constricted at the septum, fusiform, narrowly acute at both ends, straight to curved, hyaline, smooth-walled, surrounded by a gelatinous sheath. *Asexual morph*: Undetermined.

Material examined: China, Sichuan Province, Chengdu City, Jin’niu District (30°45′57″ N, 104°7′34″ E, Alt. 539 m), on dead culm of *Phyllostachys edulis*, 8 April 2021, Yicong Lv, LYC202104043 (SICAU 22-0059), living culture SICAUCC 22-0059.

Culture characteristics: Ascospores germinate in sterile water within 12 h at 25 °C. Colonies grow slowly on PDA, and reach 6 cm after 30 days at 25 °C, circular, brown to dark brown.

Notes: On the morphology, our observations were identical to the descriptions of *Seriascoma yunnanense* provided by Rathnayaka et al. [44]. Nucleotide comparisons of SSU, LSU, *tef1-α* and *rpb2* (SICAUCC 22-0059) showed high homology with the sequences of *S. yunnanense* (MFLU 19-0690), similarities are 98.37% (847/861, 0 gaps), 100% (841/841, 0 gaps), 96.59% (396/410, 0 gaps), 99.65% (855/858, 0 gaps), respectively. We report our collection as *S*. *yunnanense*.

## 4. Discussion

In this study, we confirmed seven species of saprophyte or parasitism from leaves and culms of *Phyllostachys*, corresponding to four genera. Microfungi are abundant on culms and leaves of bamboo as pointed out by Dai et al. [45]. Ascomycetes are the most abundant species on bamboo, with about 1150 taxa having been recorded [45]. Furthermore, the number of saprophytic fungi is more than that of pathogenic fungi [16,36].

The genus *Apiospora* Sacc. was recognized and described by Saccardo considering *Ap. montagnei* designated as the type species [46]. *Apiospora* has been widely accepted as a synonym for *Arthrinium* after Ellis [47]. Crous and Groenewald combined *Apiospora* species to be sexual morphs of *Arthrinium* species and synonymized under *Arthrinium* [40]. However, Pintos and Alvarado found that the morphological and ecological differences between *Apiospora* and *Arthrinium* are sufficient to support the taxonomic separation of the two genera. As a result, fifty-five species of *Arthrinium* were combined to *Apiospora* [48]. In this study, given the phylogenetic analysis with species of *Apiospora* and *Arthrinium*, in which 10 species of *Arthrinium* (*Ar.*
*a**gari, Ar. arctoscopi, Ar. fermenti, Ar. koreanum, Ar. mori, Ar. phaeospermum, Ar. pusillispermum, Ar. sargassi, Ar. taeanense, Ar. marinum*) are clustered in a well-supported clade within *Apiospora*, future studies are needed to better understand the combination of previous *Arthrinium* species with *Apiospora**. Apiospora* species have a worldwide distribution and can be found on various hosts. Most species occurred on the plants in Poaceae, although some were known from Amaranthaceae, Juncaceae, Euphorbiaceae, Cyperaceae, Restionaceae, Fagaeaeand, even seaweeds [48,49]. To date, more than 25 species have been found on bamboo, most species were saprobic on dead bamboo culms, and a few species have been reported as pathogens. For example, *Ap. arundinis* causes brown culm streak of *Phyllostachys praecox,* and *Ap. kogelbergensis* causes blight disease of *Bambusa intermedia* [16,41,50,51]. *Apiospora**. hydei*, *Ap. neosubglobosa*, and *Ap. jiangxiensis* were saprophytic on unidentified bamboo culms and leaves [41,52]. *Apiospora yunnansis* has been reported on bamboo culms of *Phyllostachys nigra* and *P. heteroclada*, which can cause bamboo blight disease of *P. heteroclada* [53,54]. In this study, four known species, *Apiospora*
*hydei*, *Ap. neosubglobosa*, *Ap. jiangxiensis,*
*and Ap. yunnansis,* were newly recorded on *Phyllostachys*
*nigra*, *P*. *heteroclada*, *P*. *bissetii,* and *P*. *aurea* respectively.

At present, *Bifusisporella* only comprises the ex-type species *B. sorghi*. In this study, we provide taxonomic details for another new species, *B. sichuanensis*, that was collected from living leaves of *Phyllostachys edulis*. *B. sorghi* was isolated as an endophyte from healthy sorghum leaves in Brazil by Silva et al. [42]. However, *B. sichuanensis* is pathogenic, causing tar spot on bamboo leaves. In addition, the sexual stage in this genus is supplemented.

Phaeosphaeriaceae is one of the most important and species-rich families in Pleosporales with diverse lifestyles [55,56], and may be found on herbaceous stems or monocotyledonous culms, branches, leaves, flowers, and woody substrates [57,58]. Currently, more than 70 genera are accommodated in Phaeosphaeriaceae [59]. Most genera in this family were introduced as monotypic genera, such as *Acericola*, *Banksiophoma*, *Bhagirathimyces, Bhatiellae, Brunneomurispora, Camarosporioides, Elongaticollum, Equiseticola, Hydeopsis, Jeremyomyces, Mauginiella, Melnikia, Neoophiobolus, Neosphaerellopsis, Neostagonosporella, Ophiobolopsis,* and *Parastagonosporella*, among others. Due to these genera being represented by a single species, resulting in few samples that could be used for taxon, the phylogenetic relationships with the related genera are sometimes not well-resolved. Based on morphological characteristics and multigene phylogeny, a novel genus, *Paralloneottiosporina,* is introduced to accommodate *Pa. sichuanensis* sp. nov. According to the field investigation, *Pa. sichuanensis* can cause leaf blight that eventually leads to leaf necrosis and plant decline in severe cases. Besides *Ph*. *violascens*, leaf blight caused by *Pa. sichuanensis* has also been observed on *P. heterocycla* and *P. tianmuensis*. This indicates that *Pa. sichuanensis* may be a common parasitic fungus on bamboos.

As only three species are accommodated within *Seriascoma*, more research is also needed for better understanding this genus [60]. *Seriascoma* is presently known as saprobic on decaying wood and dead bamboo in the terrestrial or freshwater habitats distributed in China and Thailand [16,44,61,62]. *Seriascoma. yunnanense* is found on dead branches of bamboo in Yunnan. In this study, *S. yunnanense* was saprophytic on *Phyllostachys*
*edulis*.

The previous studies have revealed a high fungal diversity associated with bamboo *Phyllostachys*. In recent years, 10 species belonging to seven genera have been described from bamboo of *Phyllostachys*, including two new genera, *Neostagonosporella* and *Parakarstenia,* established by Yang et al. on *P. heteroclada* in Sichuan Province [54,58,63,64,65,66,67,68,69]. However, the knowledge about bambusicolous fungi is incomplete and mainly remains at cataloguing stage [14]. The previous studies of identification were mostly based on morphological characteristics, and lacked molecular data. Moreover, their hosts were poorly documented or unknown [70], and specimens were absent for further re-examination. Therefore, these species need to be recollected, epitypified, and sequenced [10], and new species need to be discovered and described.

## Figures and Tables

**Figure 1 jof-08-00702-f001:**
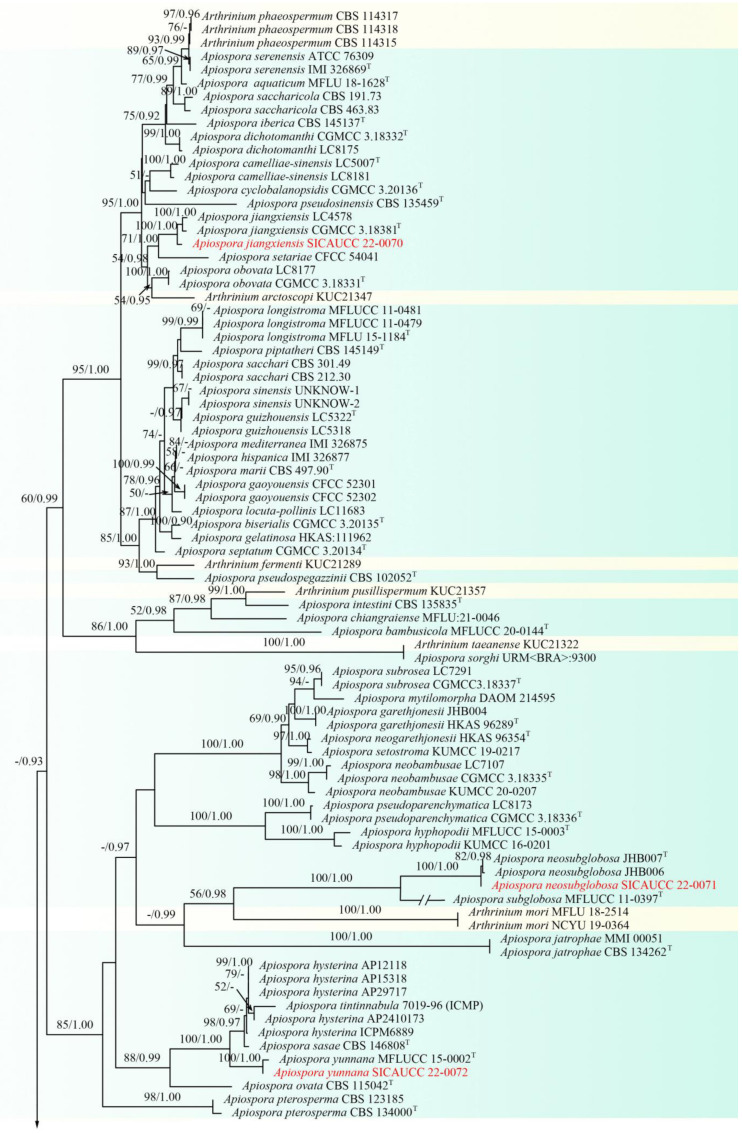
Phylogram generated from RAxML analysis based on combined ITS, LSU, *tub2,* and *tef1-α* sequence data of Apiosporaceae. Bootstrap support values for maximum likelihood (ML, left) higher than 50% and Bayesian posterior probabilities (BYPP, right) equal to or greater than 0.90 are indicated at the nodes, respectively. The sequences from ex-type strains are marked by a superscript symbol T. The newly generated sequences are written in red. *Arthrinium* species with yellow background were temporarily not combined to *Apiospora*.

**Figure 2 jof-08-00702-f002:**
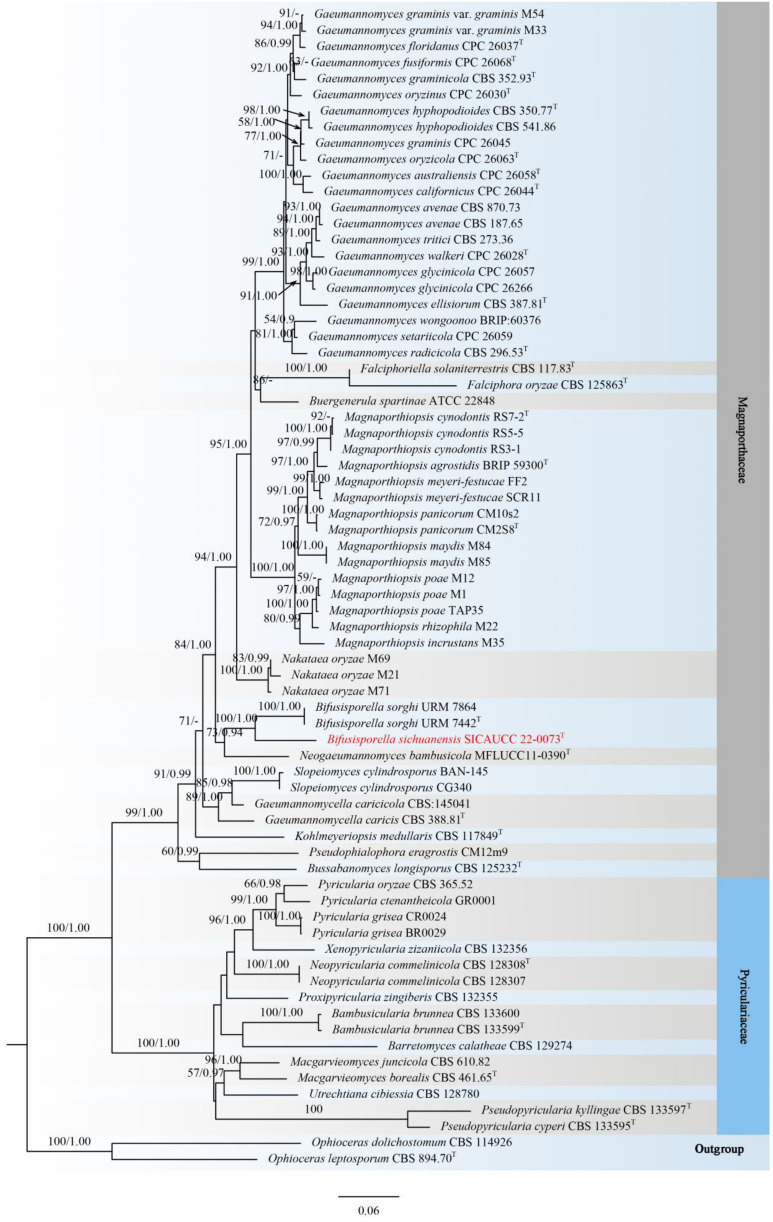
Phylogram generated from RAxML analysis based on combined ITS, LSU, *rpb1,* and *tef1-α* sequence data of Magnaporthaceae and Pyriculariaceae. Bootstrap support values for maximum likelihood (ML, left) higher than 50% and Bayesian posterior probabilities (BYPP, right) equal to or greater than 0.90 are indicated at the nodes, respectively. The sequences from ex-type strains are marked by a superscript symbol T. The newly generated sequence is written in red.

**Figure 3 jof-08-00702-f003:**
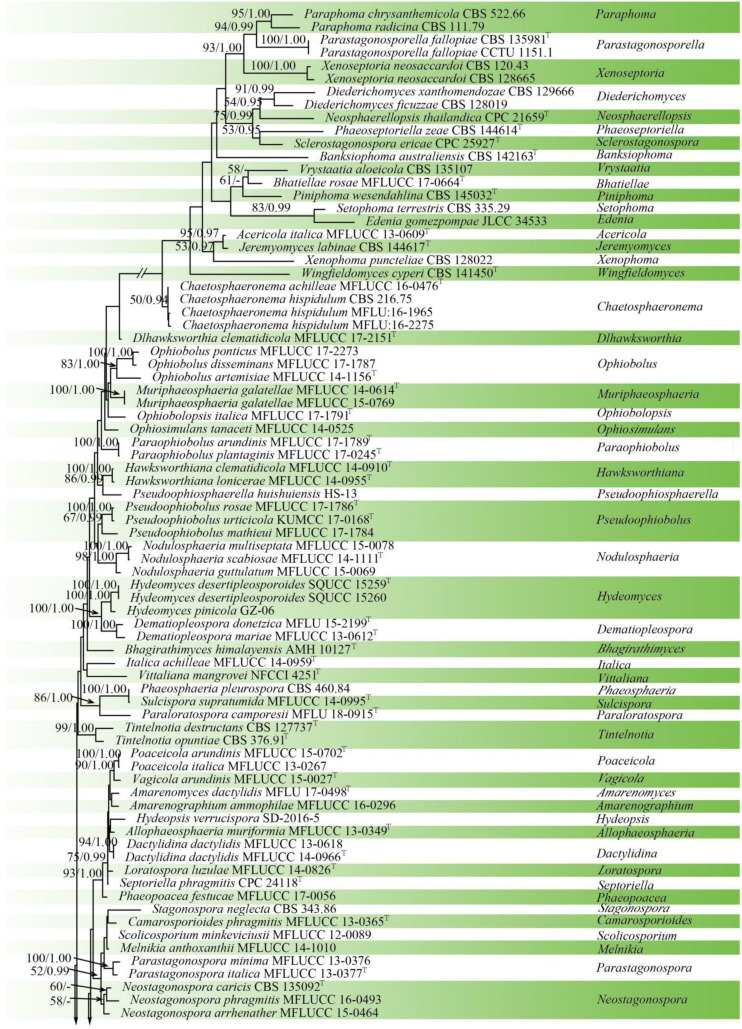
Phylogram generated from RAxML analysis based on combined ITS, LSU, SSU, and *tef1-α* sequence data of Phaeosphaeriaceae. Bootstrap support values for maximum likelihood (ML, left) higher than 50% and Bayesian posterior probabilities (BYPP, right) equal to or greater than 0.90 are indicated at the nodes, respectively. The sequences from ex-type strains are marked by a superscript symbol T. The newly generated sequences are written in red.

**Figure 4 jof-08-00702-f004:**
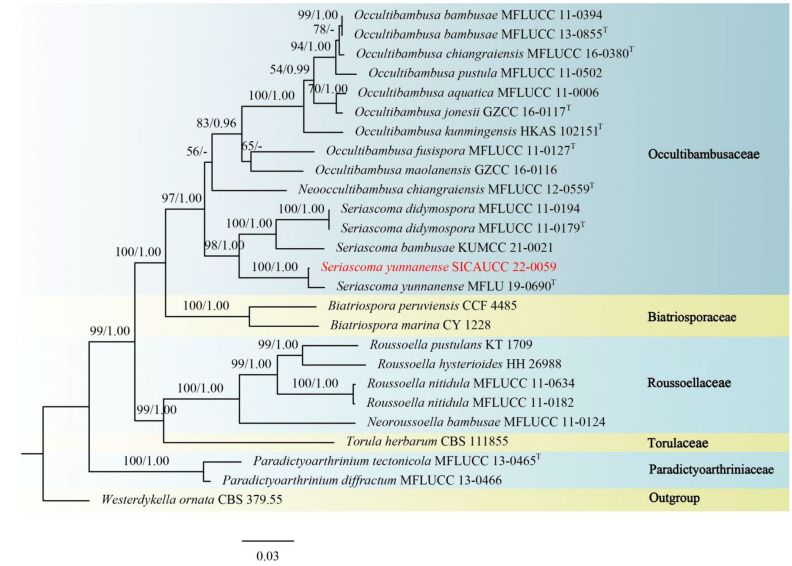
Phylogram generated from RAxML analysis based on combined ITS, LSU, *rpb**2,* and *tef1-α* sequence data of isolates within Bambusicolaceae and other representative species in Biatriosporaceae, Roussoellaceae, Torulaceae, and Paradictyoarthriniaceae. Bootstrap support values for maximum likelihood (ML, left) higher than 50% and Bayesian posterior probabilities (BYPP, right) equal to or greater than 0.90 are indicated at the nodes, respectively. The sequences from ex-type strains are marked by a superscript symbol T. The newly generated sequence is written in red.

**Figure 5 jof-08-00702-f005:**
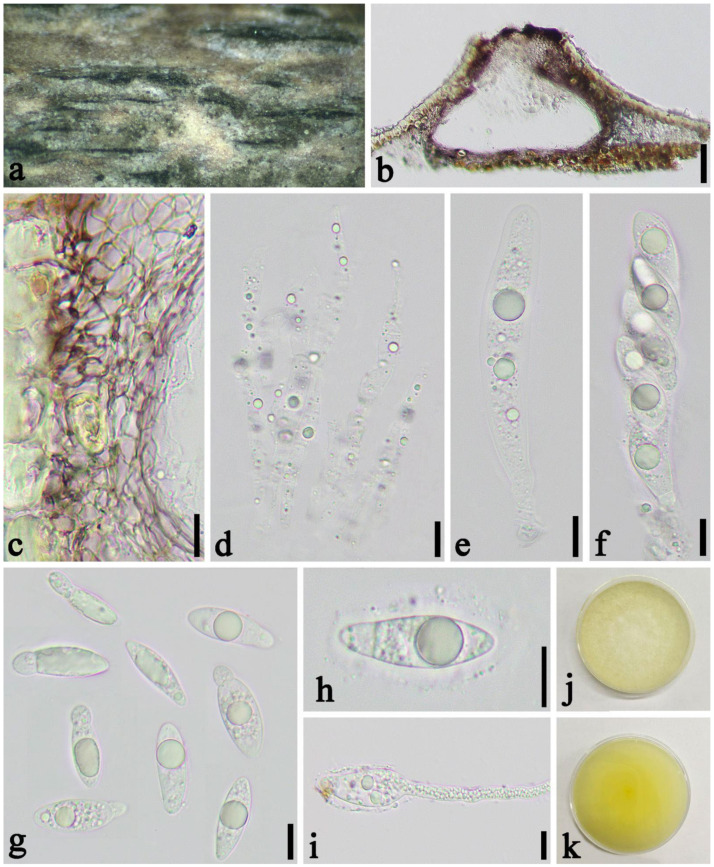
*Apiospora hydei* (SICAU 22-0032). (**a**) Ascostromata developing on bamboo branches. (**b**) Vertical sections of ascostromata. (**c**) Peridium. (**d**) Paraphyses. (**e**,**f**) Asci. (**g**,**h**) Ascospores. (**i**) Germinating ascospore. (**j**,**k**) Cultures on PDA. Scale bars: (**b**) = 50 μm, (**c**–**i**) = 10 μm.

**Figure 6 jof-08-00702-f006:**
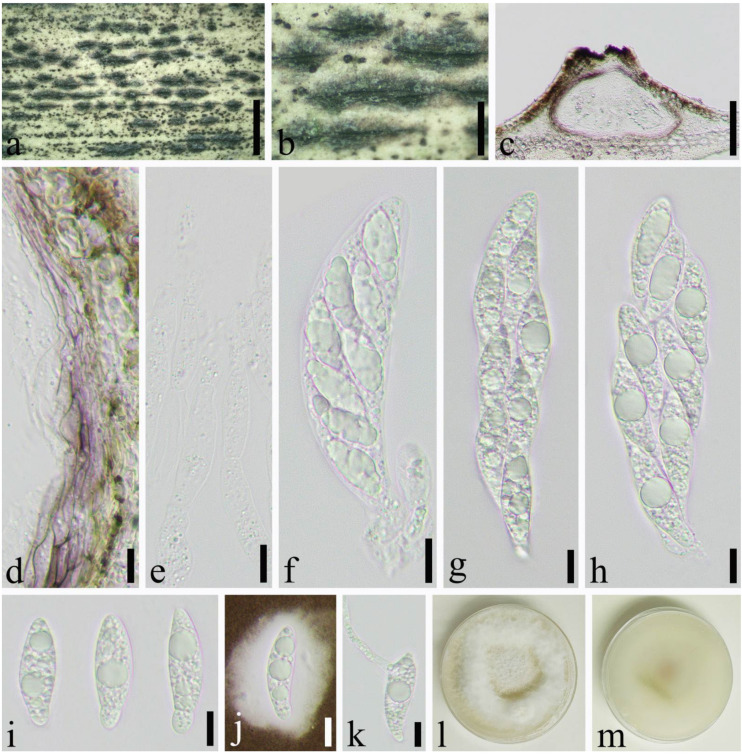
*Apiospora**jiangxiensis* (SICAU 22-0070). (**a**,**b**) Ascostromata developing on bamboo culm. (**c**) Vertical sections of ascostromata. (**d**) Peridium. (**e**) Paraphyses. (**f**–**h**) Asci. (**i**,**j**) Ascospores. (**k**) Germinating ascospore. (**l**,**m**) Cultures on PDA. Scale bars: (**a**) = 2 mm, (**b**) = 500 μm, (**c**) = 100 μm, (**d**–**k**) = 10 μm.

**Figure 7 jof-08-00702-f007:**
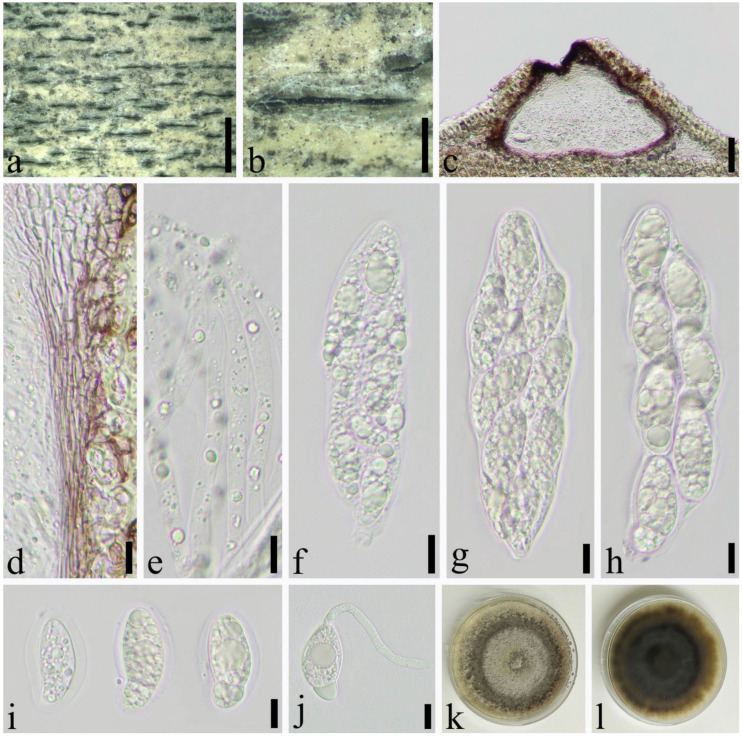
*Apiospora**neosubglobosa* (SICAU 22-0071). (**a**,**b**) Ascostromata developing on bamboo culm. (**c**) Vertical sections of ascostromata. (**d**) Peridium. (**e**) Paraphyses. (**f**–**h**) Asci. (**i**) Ascospores. (**j**) Germinating ascospore. (**k**,**l**) Cultures on PDA. Scale bars: (**a**) = 2 mm, (**b**) = 500 μm, (**c**) = 50 μm, (**d**–**j**) = 10 μm.

**Figure 8 jof-08-00702-f008:**
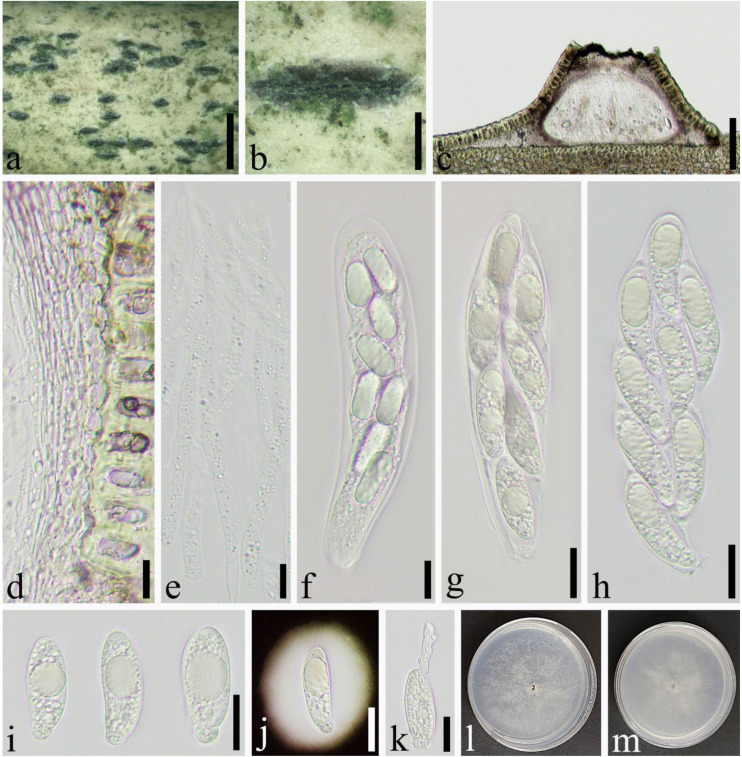
*Apiospora**yunnana* (SICAU 22-0072). (**a**,**b**) Ascostromata developing on bamboo culm. (**c**) Vertical sections of ascostromata. (**d**) Peridium. (**e**) Paraphyses. (**f**–**h**) Asci. (**i**,**j**) Ascospores. (**k**) Germinating ascospore. (**l**,**m**) Cultures on PDA. Scale bars: (**a**) = 2 mm, (**b**) = 500 μm, (**c**) = 100 μm, (**d**–**f**) = 10 μm, (**g**–**k**) = 20 μm.

**Figure 9 jof-08-00702-f009:**
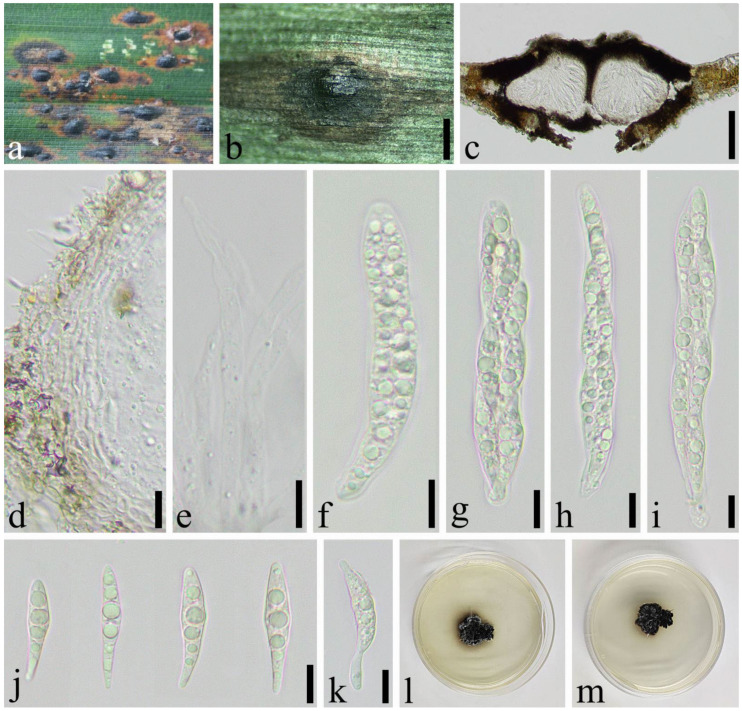
*Bifusisporella sichuanensis* (SICAU 22-0073). (**a**,**b**) Ascostromata developing on the host. (**c**) Vertical sections of ascostromata. (**d**) Peridium. (**e**) Pseudoparaphyses. (**f**–**i**) Asci. (**j**) Ascospores. (**k**) Germinating ascospore. (**l**,**m**) Cultures on PDA. Scale bars: (**b**) = 500 µm, (**c**) = 100 µm, (**d**–**k**) = 10 µm.

**Figure 10 jof-08-00702-f010:**
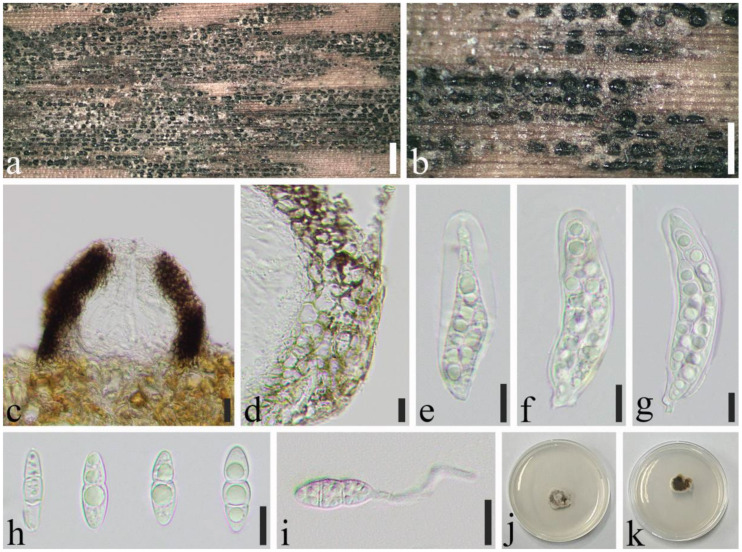
*Paralloneottiosporina sichuanensis* (SICAU 22-0074, holotype). (**a**,**b**) Ascostromata developing on the host. (**c**) Vertical sections of ascostromata. (**d**) Peridium. (**e**–**g**) Asci. (**h**) Ascospores. (**i**) Germinating ascospore. (**j**,**k**) Cultures on PDA. Scale bars: (**a**) = 1 mm, (**b**) = 500 µm, (**c**,**d**) = 20 µm, (**e**–**i**) = 10 µm.

**Figure 11 jof-08-00702-f011:**
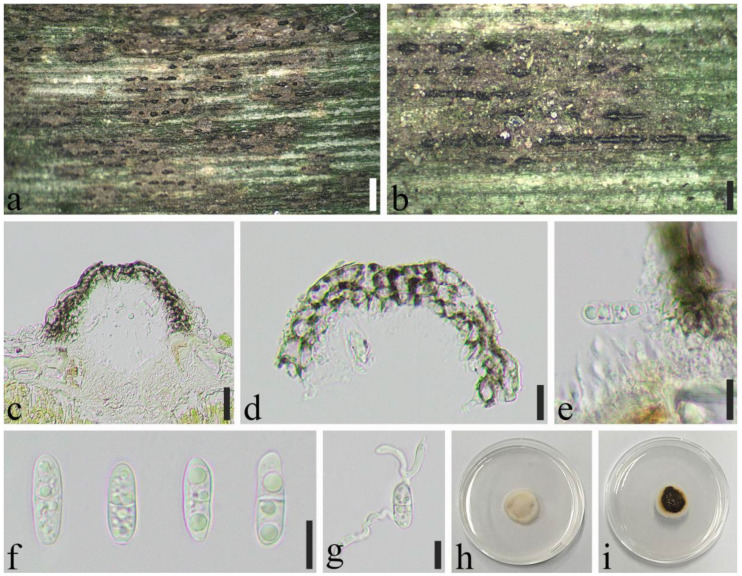
*Paralloneottiosporina sichuanensis* (SICAU 22-0075, paratype). (**a**,**b**) Conidiomata on the host. (**c**) Vertical sections of conidiomata. (**d**) Peridium. (**e**) Conidiogenous cells and developing conidia. (**f**) Conidia. (**g**) Germinating conidium. (**h**,**i**) Cultures on PDA. Scale bars: (**a**) = 500 µm, (**b**) = 200 µm, (**c**) = 20 µm, (**d**–**g**) = 10 µm.

**Figure 12 jof-08-00702-f012:**
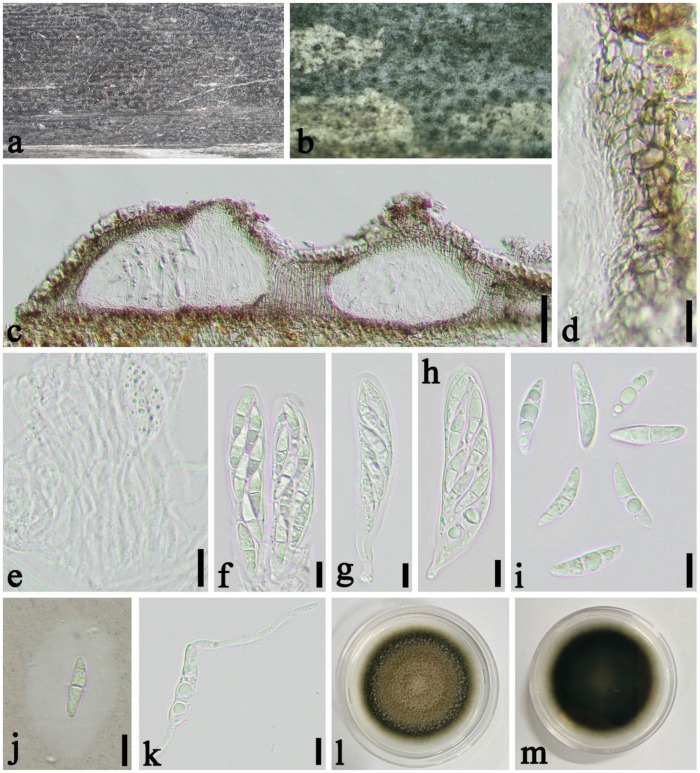
*Seriascoma yunnanense* (SICAU 22-0059). (**a**,**b**) Ascostromata developing on the host. (**c**) Vertical sections of ascostromata. (**d**) Peridium. (**e**) Pseudoparaphyses. (**f**–**h**) Asci. (**i**,**j**) Ascospores. (**k**) Germinating ascospore. (**l**,**m**) Cultures on PDA. Scale bars: (**c**) = 50 µm, (**d**–**k**) = 10 µm.

**Table 1 jof-08-00702-t001:** Selected genes for polymerase chain reaction of each genus.

Genera	Sequences Dataset
*Apiospora*	ITS, LSU, *tub2*, *tef1-α*
*Bifusisporella*	ITS, LSU, *tef1-α*, *rpb1*
*Paralloneottiosporina*	ITS, LSU, SSU, *tef1-α*
*Seriascom*	ITS, LSU, SSU, *tef1-α*, *rpb2*

## Data Availability

The datasets presented in this study can be found in the NCBI GenBank (https://www.ncbi.nlm.nih.gov/), Index Fungorum (http://www.indexfungorum.org/Names/Names.asp) (all accessed on 8 May 2022).

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
