# Peer review of "Morpho-Molecular Characterization of Microfungi Associated with Phyllostachys (Poaceae) in Sichuan, China"

_jof, 2022, doi:10.3390/jof8070702_

Round 1

Reviewer 1 Report

General comments:
- improve English grammar and style
- improve the level of scientific writing style (e.g., graceful posture)
- improve readability and clarity of the text
- improve consistency
- each idea should be presented in a different paragraph
- number until ten should be written in full (e.g., line 483: 4)

Abstract:

To simplify the text, consider removing the name of the authors that described the species for the first time (e.g., Paralloneottiosporina Q. Zeng, Y.C. Lv & C.L.).

Introduction:

Lines 26-29: Confusing text. Please consider rephrasing it.
Line27: “worldwide” in contradiction with “mainly distributed” (line 29)
Line 33: They are used in furniture, and construction
Line 34: [5, 6] and as food for human
Lines 33 – 35: Please be consistent. If examples were given for construction applications, the same should be done for the other applications.
Line 39: pathogens fungi, a large number of saprobic and endophytic fungi
Line 41: place the references accordingly
Lines43 – 51: Please consider simplifying the text to improve readability
Line 52: s fungi with bamboo species are is not clear

Material and Methods:

Line 61: the specimens were collected
Line 62: taken back to the laboratory
Lines 63-65: Confusing text. Please consider rephrasing it.
Line 79: Please confirm culture duration (30 days?).
Lines: 79 – 87: Please consider simplifying the text to improve readability and some references are missing. Examples: The studied genomic regions were internal transcribed spacers (ITS) (primers ITS5/ITS4) [ref], the partial large subunit nuclear rDNA (LSU) (primers NS1/NS4) [26], …
Line 92: Please consider briefly describing the sequencing procedures.
Lines 94 - 98: Please consider simplifying the text to improve readability. Transform data into a table.

Results:

Line 111: Please consider replacing the word “character” for a more suitable one. Apply this suggestion throughout the text.
Lines 202 – 213: Please consider abbreviating the name of the bamboo species (e.g., Ph. nigra). Why are the words “Ascostromata”, “Peridium”, “Hamathecium”, "Ascospores", and "Asexual morph" written in italics? Apply these suggestions throughout the text.

Lines 373 – 375: Please consider changing the data to Homology percentage to facilitate the interpretation by the reader.

Discussion:

Line 482: and culms of Phyllostachys, representing 4 corresponding to four genera.
Lines 486 – 487: The genus Apiospora Sacc. was recognized and established described by Saccardo with considering Ap. montagnei designated as the type species [46].
Line 523: Parastagonosporella et.al. among others
Line 528: Besides Phyllostachys Ph. praecox,
Line 534: S. Seriascoma yunnanense is found on

Figures:

Line 133: The newly generated sequences written are in red. Apply this suggestion throughout the text.

Author Response

Thank you for your comments and suggestions. I have made corresponding changes in the paper, marked with bright yellow. And answer and explain some questions.

  1. Lines: 79 – 87: Please consider simplifying the text to improve readability.

Selected genes and primers for polymerase chain reaction of each genus need to be described in detail and cannot be simplified.

  1. Line 111: Please consider replacing the word “character” for a more suitable one. Apply this suggestion throughout the text.

The word "character" is used in reference to previous literature, and no better replacement has been found, so it has not been modified.

  1. Why are the words “Ascostromata”, “Peridium”, “Hamathecium”, "Ascospores", and "Asexual morph" written in italics? Apply these suggestions throughout the text.

In order to quickly find the description of the corresponding morphological structure.

  1. Lines 373 – 375: Please consider changing the data to Homology percentage to facilitate the interpretation by the reader.

Bifusisporella sichuanensis and B. sorghi are different species. In addition to morphological differences, base-pair differences are needed to support this result. Therefore, we cannot change the data to Homology percentage.

Reviewer 2 Report

The manuscript articleMorpho-Molecular Characterization of Microfungi Associated 2 with Phyllostachys (Poaceae) in Sichuan, China it is scientifically well written, however we emphasize that we are not qualified to review English. We found only one error in the material and methods "Line 103: Phylogenetic tree: the usual in phylogeny is phylogram. The suggestion is to use the same spelling that appears in line 130 Figure 1.

Author Response

Thank you for your comments and suggestions. I have made corresponding changes in the paper, marked with bright yellow.

Reviewer 3 Report

Dear authors,

This manuscript "Morpho-Molecular Characterization of Microfungi Associated with Phyllostachys (Poaceae) in Sichuan, China" by Zenf et al. is an interesting piece of work and described new/known species based on morphological and molecular features. Please see my comments and suggestions in the attached file and please carefully check unnecessary words and correct the reference style in line with the journal guidelines.

Best wish

Author Response

Thank you for your comments and suggestions. I have made corresponding changes in the paper, marked with bright yellow. And answer and explain some questions.

  1. Better to add phylogeny tree and follow by photoplates of each genera.

The journal requires insert the graphics ( schemes, figures, etc. ) in the main text after the paragraph of its first citation.

  1. Why is only the genus Bifusisporelladescribed in detail ?

At present, Bifusisporella comprises only the ex-type species B. sorghi, and no records on its sexual morph. That there is a need to amend the morphological circumscriptions of the genus.

  1. Discussion:Better move to notes of Apiospora and Phaeosphaeriaceae.

'Notes' only compared morphologically similar and phylogenetically related species. Apiospora and Phaeosphaeriaceae are not described and discussed. So this part cannot be moved to notes.

Round 2

Reviewer 3 Report

Dear authours,

Thank you for the revised manuscript.

I have a few suggestion in the attached file

Kind regards

Author Response

Thank you for your comments and suggestions. I have made corresponding changes in the paper, marked with bright yellow. High-resolution Figures1-4 have been uploaded as attachments.